# Genetic Variation among Pharmacogenes in the Sardinian Population

**DOI:** 10.3390/ijms231710058

**Published:** 2022-09-02

**Authors:** Maria Laura Idda, Magdalena Zoledziewska, Silvana Anna Maria Urru, Gregory McInnes, Alice Bilotta, Viola Nuvoli, Valeria Lodde, Sandro Orrù, David Schlessinger, Francesco Cucca, Matteo Floris

**Affiliations:** 1Institute for Genetic and Biomedical Research, National Research Council, 07100 Sassari, Italy; 2Institute for Genetic and Biomedical Research, National Research Council, 09042 Monserrato, Italy; 3Hospital Pharmacy Unit, Trento General Hospital, Autonomous Province of Trento, 38122 Trento, Italy; 4Department of Chemistry and Pharmacy, School of Hospital Pharmacy, University of Sassari, 07100 Sassari, Italy; 5Biomedical Informatics Training Program, Stanford University, Stanford, CA 94305, USA; 6Department of Biomedical Sciences, School of Medicine, University of Sassari, 07100 Sassari, Italy; 7Department of Biomedical Sciences, University of Sassari, 07100 Sassari, Italy; 8Medical Genetics, Department of Medical Sciences and Public Health, University of Cagliari, 09126 Cagliari, Italy; 9Laboratory of Genetics and Genomics, National Institute on Aging, National Institutes of Health, Baltimore, MD 21224, USA

**Keywords:** pharmacogenetics, pharmacogenomics, drug response

## Abstract

Pharmacogenetics (PGx) aims to identify the genetic factors that determine inter-individual differences in response to drug treatment maximizing efficacy while decreasing the risk of adverse events. Estimating the prevalence of PGx variants involved in drug response, is a critical preparatory step for large-scale implementation of a personalized medicine program in a target population. Here, we profiled pharmacogenetic variation in fourteen clinically relevant genes in a representative sample set of 1577 unrelated sequenced Sardinians, an ancient island population that accounts for genetic variation in Europe as a whole, and, at the same time is enriched in genetic variants that are very rare elsewhere. To this end, we used PGxPOP, a PGx allele caller based on the guidelines created by the Clinical Pharmacogenetics Implementation Consortium (CPIC), to identify the main phenotypes associated with the PGx alleles most represented in Sardinians. We estimated that 99.43% of Sardinian individuals might potentially respond atypically to at least one drug, that on average each individual is expected to have an abnormal response to about 17 drugs, and that for 27 drugs the fraction of the population at risk of atypical responses to therapy is more than 40%. Finally, we identified 174 pharmacogenetic variants for which the minor allele frequency was at least 10% higher among Sardinians as compared to other European populations, a fact that may contribute to substantial interpopulation variability in drug response phenotypes. This study provides baseline information for further large-scale pharmacogenomic investigations in the Sardinian population and underlines the importance of PGx characterization of diverse European populations, such as Sardinians.

## 1. Introduction

Drug treatments are characterized by substantial difference in terms of efficacy and/or safety in different patients. Adverse drug reactions (ADR), including allergic, pseudo-allergic, and exaggerated pharmacological reactions to medications, are a relatively common result of drug treatment, accounting for at least 5% of hospital admissions, with an overall fatality of 0.15% and an annual cost of >500 M$, only for the UK National Health Service [1,2]. These data highlight the social and economic costs of ADRs and the urgent need to find effective strategies to ameliorate drug efficacy and reduce ADR.

The same drug, once other parameters are fixed, can have different therapeutic effects in different people due to causal genetic variants [3]. The analysis of the genetic variability modulating the individual’s drug response (pharmacogenetics, PGx) has, thus, received great attention for its capacity to provide a new way to optimize drug therapies in terms of optimal dosing to improve drug efficacy and reduce toxicity risk [4]. As a result, a patient may receive the right drug at the right dose the first time they consult their doctors such that efficacy is guaranteed, and the risk of ADR is reduced. From a pharmaceutical point of view, PGx variants can influence pharmacokinetics and pharmacodynamics drugs, thus influencing dosing, formulation sensitivity and drug-hypersensitivity reactions.

An individual’s drug response can be assessed through the identification, by genotyping arrays or sequencing, of well-characterized genetic variants and specific haplotypes in key genes implicated in drug processing. For example, the gene CYP2D6 is characterized by the presence of over 100 haplotypes, which share SNPs and include gene duplications and deletions, strongly influencing the metabolism and/or bioactivation of many clinically used drugs and, thus, determining a phenotype. In this example, phenotypes are assigned to haplotypes that contains specific and relevant SNPs to differentiate CYPD6 functions [5].

The interest in ameliorating drug efficacy, while reducing ADR, promotes the development of tools to properly analyze the correlation between variability in the genome and individual’s drug response. For example, The Pharmacogenomics Knowledge Base (PharmGKB: http://www.pharmgkb.org (accessed on 22 March 2022) [6,7]) covers much information about pharmacogenomics and provides a convenient approach for researchers. The Pharmacogenetics of Membrane Transporters (PMT) database is another tool focused on the effect of genetic variation in the response to drugs that interact with membrane transport proteins [8,9].

Furthermore, the increasing availability of accurate classifications of pharmacogenetic variants and haplotypes, together with guidelines for their clinical translatability, allow analysis of the potential impact of pharmacogenetics programs in many populations for which large-scale genomic resources exist [10,11]. The analysis of the prevalence of PGx-risk variants in target populations, in combination with actual data on drug usage, make it possible to predict the proportion of the population for which genetics could lead therapy decision. Overall, the following axes could support a coordinate pharmacogenetic program in the European healthcare systems: (i) the analysis of PGx variant prevalence, (ii) the results of clinical trials evaluating patient outcomes and cost-effectiveness of PGx-markers [12] and (iii) outcomes of implementation strategies [13,14].

Sardinians, a population for which large-scale genomic data are available, is particularly well suited for genetic studies. Sardinians are the contemporary human population that has retained the highest degree of inheritance from early European farmers who lived in the Neolithic period along with significant ancestry from western hunter gatherers who lived in the late Paleolithic period [15,16]. This is due to founder effects during the initial settlement of the island and the scarcity of gene flow from other populations during later periods [15,16]. As a result of its past evolutionary history the Sardinians are now a reservoir of ancient European genetic variants that are currently very rare elsewhere and may have relevant clinical consequences [17,18,19,20]. Genetic factors and the distinct genetic structure of the Sardinians thus present an excellent opportunity to also look for new pharmacogenetic information.

Here, we profiled pharmacogenetic variation in fourteen clinically relevant genes in 1577 unrelated sequenced Sardinians. We used PGxPOP [21], a PGx allele caller, based on the guidelines created by the Clinical Pharmacogenetics Implementation Consortium (CPIC), to identify the main phenotypes associated with the PGx alleles most represented in Sardinians. We estimated that 99.43% of Sardinian individuals might potentially respond atypically to at least one drug, and that, on average, each individual is expected to have an abnormal response to about 17 drugs. Furthermore, we highlighted differences in haplotype and diplotype frequencies of star alleles as compared to other populations and estimated that for 27 drugs the fraction of the population at risk of atypical responses to therapy is more than 40%. These findings represent the foundation for further large-scale and more detailed pharmacogenomic investigations in Sardinia, and, at the same time, underline the importance of the pharmacogenomic characterization of ethnically diverse European populations, as exemplified by Sardinians.

## 2. Results

### 2.1. Haplotype and Phenotype Calling

To identify clinically relevant pharmacogenetic variation in 14 important genes for which Clinical Pharmacogenetics Implementation Consortium (CPIC) has created detailed gene/drug clinical practice guidelines, we processed the genomic sequence data from our SardiNIA cohort (1577 individuals) with PGxPOP [21].

PGxPOP is a PGx matching engine that is based on PharmCAT and uses its PGx allele definitions to characterize PGx haplotype, diplotype and phenotype frequencies. It extends the capabilities of PharmCAT by generating diplotypes from population scale datasets [22]. In the analyzed Sardinia cohort, 99.43% of the 1577 participants carried at least one diplotype associated with a predicted non-typical response phenotype across the 14 pharmacogenes analyzed (Table 1). Furthermore, for each participant, we were able to predict an average of about 4 phenotypes of non-typical drug response (Mean = 3.44, Min = 1, Max = 8, Figure 1). These numbers were in line with what has been observed in the UK Biobank cohort, where participants were previously reported to carry on average 3.7 nontypical response diplotypes for the 14 pharmacogenes, with 99.5% of participants carrying at least 1 nontypical drug response diplotype [21].

Although, in general, we observed some agreement with the observations in UKBB, we noticed some differences in terms of haplotype and diplotype frequencies. Largest absolute discordance in star allele frequencies (delta MAF ≥ 10%) between Sardinians and individuals with European ancestry of the UK Biobank cohort were observed for CYP2D6*1 and *119 (results that can be affected by the fact that structural variants were not called in this analysis), CYP4F2 alleles *1, *2 (this being extremely rare in Sardinians) and *2 + *3, VKORC1 alleles −1639A and −1639G, and for SLCO1B1 alleles *1A and *14 (Appendix A).

In addition, out of 133 diplotypes called by PGxPOP for the 14 genes analyzed, only 13 diplotypes (10%) had an absolute difference in frequency ranging from 10% to 25%, as compared to UKBB European individuals (Appendix A). We detected a total of 22 different phenotypes for the 14 pharmacogenes, but for *CYP2D6* and *UGT1A1* our analysis did not detect any non-typical drug response diplotype in the SardiNIA cohort (Table 2). To understand whether the unexpected result for *CYP2D6* was due to a limitation of the SardiNIA genetic map in the *CYP2D6* region, we performed an exploratory analysis with 65 deeply sequenced samples from the same cohort (mean coverage > 30×, data not shown here), where *CYP2D6* star alleles were called with the Aldy tool [23]. The results showed that common *CYP2D6* star alleles had comparable frequencies to those reported for the European samples in the UKBiobank cohort [21].

Based on these analyses, we could predict that, for at least 27 drugs, about one third of the population was at risk of an atypical response (Table 3) and that, on average, each individual was expected to have an abnormal response to about 17 drugs (Table 1).

The genes with the highest number of non-typical phenotypes were *VKORC1* (N = 1192; 76%), *CYP2C19* (N = 862; 55%) and *IFNL3* (N = 856; 54%). Genes with the most ‘not available’ phenotypes were *CYP2D6* (47.43% of subjects), *UGT1A1* (48.31%), *CYP4F2* (48.45%) and *SLCO1B1* (12.75%). Alleles with unknown or uncertain function, leading to an “indeterminate” phenotype, were found in 6 genes (*CYP2B6*, *CYP2D6*, *CYP4F2*, *SLCO1B1*, *TPMT* and *UGT1A1*).

If we focused on the frequency of non-typical drug response phenotypes in the Sardinian cohort compared to the European sub-population of UK Biobank, the more frequent in Sardinians were “Decreased warfarin dose” for the *VKORC1* gene (26% vs. 14%) and “Intermediate metabolism” for the *CYP3A5* gene (19% vs. 13%). For 5 other non-typical drug response phenotypes, the difference in frequency was between 1 and 5%, while for all other non-typical phenotypes the frequency in the Sardinian population was always lower than in the European UK Biobank cohort (full details are available in Appendix A).

### 2.2. Actionable Pharmacogenomic Variants in Sardinian Genomes

We extended the above analysis by considering the carrier status of clinically actionable variants (PharmGKB level 1A/1B) in each of the 1577 Sardinian unrelated individuals.

Among the 3073 clinical annotations in the PharmGKB database (accessed on 22 March 2022), 141 single-nucleotide variants (SNVs) and 132 haplotype variants had the highest level of evidence (1A/1B or 2A/2B), while 2652 SNVs and 247 haplotype variants had a lower level (3 or 4). The SNV variants were then overlapped with the variants in the Sardinian population and their prevalence was evaluated based on their allele frequencies in both a dataset of 1577 unrelated Sardinians and in the gnomAD dataset (non-Finnish Europeans), which established extensive interpopulation differences. In this analysis, we considered only variants in non-cytochrome genes, yielding results as follows.

Among the variants associated with at least one of the highest levels of evidence, we identified, in our analysis, 13 variants for which the absolute difference in allele frequency between Sardinians and Europeans was at least 5%, of which 5 had a MAF in Sardinia that was at least 10% higher than the Europeans (Table 4 for a summary, Table 5 for clinical annotations). These 13 high-evidence pharmacogenetic variants were involved—according to 26 PharmGKB clinical annotations—in the response to the following 9 different drugs: the anticoagulants Acenocumarol, Penprocoumon, and Warfarin; the cholesterol-lowering agent Pravastatin; the anticancer agents Capecitabine, Fluorouracil; and the immunosuppressants Etanercept, Methotrexate, and Rituximab. Many of the variant-drug pairs (2 out of 3) were relevant for dosage and efficacy of the analyzed drug, while the remaining were relevant for toxicity (Table 5). Interestingly, about half of these differentiated pharmacogenetic variants with high priority level were relevant for drugs directed to pediatric populations. In more detail, 4 variants in 4 genes (*VKORC1*, *CYP4F2*, *DPYD* and *MTHFR*) were marked as important for pediatric prescription of 4 drugs (Phenprocoumon, Warfarin, Fluorouracil and Methotrexate). For 4 of these variants, we had at least one alert with Evidence level 1A or 1B (rs2108622, rs9923231 and rs9934438, whose minor allele was about 10% more frequent in Sardinians than in Europeans, whereas the rs1801265 minor allele was about 6% less frequent in Sardinians) (Table 5). It should be noted that SNPs rs9934438 and rs9923231 are in high LD (r2 = 1) in both Sardinians and Europeans.

Among the other pharmacogenetic variants with lower levels of evidence (3 and 4), we identified 169 variants for which the minor allele frequency was at least 10% higher in Sardinians compared to Europeans, and a further 72 variants with an opposite trend (Table 6 for a summary, Appendix Afor clinical annotations). The 169 low-evidence variants were involved in the response to 201 different drugs (a total of 405 variant-drug pairs). Most variant-drug pairs were relevant for drug efficacy (39% of annotations) and for toxicity (37% of annotations) (Appendix A). Overall, 41 of the variant-drug pairs (about 10%) were relevant for drugs directed to the pediatric population.

For another 110 variant-drug pairs with lower levels of evidence (5 of which were relevant for the pediatric population), 72 unique variants, for which the minor allele frequency was at least 10% lower in Sardinians compared to Europeans, were involved in the response to 91 different drugs.

## 3. Discussion

We estimated the potential impact of the large-scale introduction of pharmacogenetic practices in the Sardinian population by evaluating the prevalence of clinically relevant pharmacogenetic variants in a core set of 1577 unrelated sequenced individuals, representative of the entire population (Sardinia has 1.5 M residents on the island and a similar number of individuals of Sardinian descent spread across the world). To this end, we used PGxPOP [21], a PGx matching engine that is based on PharmCAT and uses its PGx allele definitions, to characterize PGx allele and phenotype frequencies. Using this analysis, it was possible to estimate the theoretical number of Sardinian individuals exposed to adverse reactions to a range of drugs. In more detail, the frequencies of two phenotypes (“Decreased warfarin dose” and “Possibly decreased warfarin dose”) involving warfarin, a widely used anticoagulant drug, were among the most interesting findings from this analysis. The two atypical phenotypes are determined by diplotypes of the *VKORC1* gene [24] and affected a total of 1192 individuals in our cohort (i.e., about 3 of 4 individuals). Overall, common genetic variants in this gene, but also in *CYP2C9*, *CYP4F2*, and the CYP2C cluster (e.g., rs12777823), plus known nongenetic factors, account for 50% of warfarin dose variability [24].

Other phenotypes potentially affecting a large proportion of the population were the “Intermediate Metabolizer” and “Poor Metabolizer” phenotypes, which are determined by cytochrome *CYP2C9* diplotypes and affected a total of 650 individuals in our cohort (about 41% of the cohort analyzed). These phenotypes have important effects on the ADME-Tox of Nonsteroidal Anti-Inflammatory Drugs, such as celecoxib, flurbiprofen, lornoxicam, and ibuprofen. According to CPIC guidelines [25], the diplotypes involved may result in a higher-than-normal risk of adverse events, especially in individuals with other factors affecting clearance of these drugs, such as hepatic impairment or advanced age. The same guidelines suggest a reduced dosage of these drugs and monitoring of adverse effects. The same cautions can be extended to other drugs, such as meloxicam, piroxicam and tenoxicam.

An important finding concerned two atypical phenotypes related to the *SLCO1B1* gene (“Decreased Function”, N = 324 individuals, and “Poor function”, N = 51), which globally affect almost 1 in 4 individuals, and are important for the metabolism of important drugs, such as Atorvastatin (second among the top thirty active drugs both for consumption and expenditure in Italy) [26], and Fluvastatin, Lovastatin, Pitavastatin, Pravastatin, Rosuvastatin and Simvastatin. According to CPIC guidelines [27], these phenotypes can impact the starting dose and suggest an adjustment of doses based on disease-specific guidelines. According to suggestions in the same guidelines, prescribers should be aware of possible increased risk for myopathy.

We could then hypothesize that an important impact on the frequency of adverse effects could be caused by the high diffusion of atypical phenotypes (“Intermediate metabolizer”, N = 450, “Poor metabolizer”, N = 32, “Rapid metabolizer”, N = 343 and “Ultrarapid Metabolizer”, N = 37) attributable to diplotypes of the gene *CYP2C19*, involved in the metabolism of some of the most widely used antidepressants in Italy, including escitalopram and sertralin. According to the guidelines [28], among the problems caused by an incorrect dosage are increased risk for adverse cardiac and cerebrovascular events.

Special attention should be paid to the 103 individuals (approximately 6.5%) who are at high risk of severe toxicity due to antineoplastic drugs, such as azathioprine, mercaptopurine, and thioguanine, because of atypical phenotypes determined by diplotypes of the *TPMT* gene.

In a second phase of analysis, we aimed to identify the variants of pharmacogenetic interest that were more differentiated in Sardinia than in the general European population (taking as reference the genetic data of gnomAD version 2.1). In this analysis, we distinguished highly relevant PGx variants (levels of evidence 1A, 1B, 2A and 2B) from those of lower relevance (levels 3 and 4).

The strongest difference in terms of allele frequency was seen for the rs396991 variant located in the *FCGR3A* gene and which could be relevant for patients treated with Rituximab, according to a 2B level of evidence documented by PharmGKB [29]. In fact, the C allele of the rs396991 variant had a frequency 1.5 times higher in Sardinia (AF = 0.528) than in the rest of Europe (AF = 0.344) This difference may significantly affect the efficacy of Rituximab, used in the treatment of certain types of cancer and autoimmune disorders, including Rheumatoid Arthritis and Neuromyelitis Optica. Indeed, patients with a CC genotype may have an increased response to the drug compared to patients with AA and AC genotypes.

Another variant of special interest was rs8050894, for which the frequency of the G allele was 1.34 times higher in Sardinia (AF = 0.523) than in the general European population (AF = 0.389). This variant has a role, supported on a type 1B level of evidence, in influencing warfarin dosage. According to the guidelines, patients with the GG genotype may require a lower dose of warfarin as compared to patients with the CC genotype. The variant is part of a haplotype of variants in the *VKORC1* gene, all of which are associated with warfarin dosing. Among them, the one with the strongest level of evidence was rs9923231, whose T allele was 1.309 times more frequent in Sardinians (AF = 0.509 versus 0.389). This last variant was also relevant for the pediatric population and had relevance not only for the dosage of Warfarin, Acenocoumarol and Phenprocoumon, but also for the resultant efficacy and toxicity of these drugs. Of note, the genotypes of *VKORC1*-1639G > A (rs9923231) are mentioned in the FDA Label of Warfarin.

Warfarin inhibits *VKORC1* to prevent regeneration of a reduced form of vitamin K necessary for clotting factor activation [30]. The common variants, noted in our analysis, are located in the 5′UTR and introns of the *VKORC1* gene and are associated with reduced gene expression and related effects on warfarin dosage. Warfarin and Acenocoumarol are common oral anticoagulant prescribed for the treatment and prevention of thromboembolic events for which genetic variants in several genes (*CALU*, calumenin; *CYP*, cytochrome P450 family members; *GGCX*, gamma-glutamyl carboxylase; *NQO1*, NAD(P)H quinone dehydrogenase 1; *VKORC1*, vitamin K epoxide reductase) have been associated with the need for carefully calibrated dosage to prevent bleeding episodes.

The influence of *VKORC1* polymorphisms on vitamin K antagonist dose requirements provides a remarkable example of pharmacogenomic diversity worldwide. This is documented by the International Warfarin Pharmacogenomic Consortium (IWPC) datasets, comprising 5700–6200 patients recruited from four continents, and ascribed to three ‘racial’ groups, namely Asians, Blacks (mainly African Americans) and Whites [31].

Furthermore, considering the 19 HLA alleles associated with adverse events to the therapy with the highest level of evidence, mention should be made of HLA-B*58:01, which has been shown to have a strong effect on the development of severe cutaneous adverse reactions (SCARs), including Stevens—Johnson syndrome and toxic epidermal necrolysis after treatment with allopurinol, the common treatment for hyperuricemia and gout. However, the frequency of HLA-B*58:01 significantly differs between different ethnic groups. The frequency of HLA-B*5801 is the highest in Han-Chinese (20%), Korean (12%), and Thai (13%), but is much less frequent in Japanese (0.1%) The same allele, however, is also much more frequent in Sardinians (11%) than in other European populations (France 1.5%) [32]. We believe that this evidence is of particular relevance, given that Sardinia has the highest percentage of reports of adverse events [26] following allopurinol administration of the total number of adverse reports registered in Italy (1.9% compared to an average of 0.41% in the other Italian regions).

Among the most differentiated variants with lower levels of evidence, two independent variants rs3815087 (allele A) and rs3131003 (allele A) located in *PSORS1C1* region, were of particular interest. Both variants were highly frequent in Sardinians compared to European populations (delta frequency > 29%), have been associated with epidermal necrolysis and Stevens-Johnson syndrome [33,34] after allopurinol therapy (evidence levels 3 and 4, respectively) and show coincident, strong association with psoriasis (*p* = 1.2 × 10^−294^, OR = 2.93; *p* = 1.4 × 10^−105^, OR = 1.64) [https://genetics.opentargets.org (accessed on 22 March 2022); rs3815087 and rs3131003 variants respectively]. They were very common in Sardinia (AR 50% and 74.6%), and, thus, screening for these variants before therapy could be important. It is, thus, not surprising that the variant rs2233945, localized in the same *PSORS1C1* gene, modulates the response to etanercept, a TNF inhibitor used for psoriasis and other autoimmune disorders, including rheumatoid arthritis. Allele A in that locus has been associated with increased etanercept efficacy in comparison to allele C: and at the same time allele A has been associated with protection from psoriasis. This variant has been in linkage disequilibrium with one canonically described for allopurinol adverse events, rs9263726, the variant tag for HLA-B*58:01.

## 4. Material and Methods

### 4.1. Study Population and Data Sets

#### 4.1.1. Dataset

In this study we focused on a subset of 1577 unrelated sequenced Sardinian samples from the Sardinian sequence-based reference panel. These were mainly the unrelated parents of a larger sample set of 3514 individuals (mainly trios) sequenced at low coverage (average coverage 4.2×), which also included their children. The sample set included 2090 individuals belonging to the SardiNIA cohort study in the subregion of Ogliastra and 1424 individuals deriving from a case-control study on autoimmunity collected across the Island [16,19]. All participants signed informed consent to study protocols approved by the Sardinian Regional Ethics Committee (protocol no. 2171/CE).

#### 4.1.2. Genotyping and Imputation

All the genetic analyses were performed using a genetic map based on 6602 samples genotyped with 4 Illumina arrays (OmniExpress, ImmunoChip, Cardio-MetaboChip and ExomeChip), as previously described [19]. Imputation was performed on a genome-wide scale using a Sardinian sequence-based reference panel of 3514 individuals and the software Minimac3 on pre-phased genotypes. After imputation, only markers with imputation quality (RSQR) > 0.3 for estimated minor allele frequency (MAF) ≥ 1% or > 0.6 for MAF < 1% were retained for further analyses, yielding ~22 million variants (20,143,392 SNPs and 1,688,858 indels).

### 4.2. Variant Calling and Relatedness

Variant calling and phase assignment procedures leveraged the family structure of the extended sample of 3514 individuals. Allele frequencies were estimated from the subset of 1577 unrelated individuals and compared to the corresponding frequencies available in gnomAD project v2 (European non-Finnish subset) [35].

Relatedness was estimated by computing the genome-wide proportion of pairwise IBD (π) on a random set of 1 million SNPs with an MAF > 0.05 in 1000 Genomes populations (Phase 3 v5) [https://www.internationalgenome.org/home (accessed on 1 May 2019)]. Starting from the dataset of 3514 individuals, for each pair of individuals with π > 0.05, we preferentially removed the offspring if in a trio; otherwise, we removed the individual with the larger summed value of π across all other relationships with π > 0.05. Once all removals were completed, we had a total of 1577 samples for the allele frequency analyses.

### 4.3. Pharmacogenetics Resources

#### 4.3.1. Haplotype and Phenotype Calling

We used PGxPOP [https://github.com/PharmGKB/PGxPOP (accessed on 1 December 2021)] [21], to calculate specific haplotype, diplotype, and phenotype frequencies for fourteen genes in our multisample VCF file. PGxPOP uses the PharmCAT allele definition files [https://github.com/PharmGKB/PharmCAT (accessed on 22 March 2022)] [22] and reports exact matches to the allele definitions based on the provided phased genotype data.

Gene phenotypes were determined to have a non-typical response if any CPIC guidance recommended an alternate dosage or drug for that phenotype, as defined by McInnes et al. [21] (Appendix A).

Currently, the CPIC guidelines (https://cpicpgx.org/genes-drugs/, accessed on 14 July 2022) indicate a set of 64 recommendations for the 14 genes profiled by PGxPOP, for a total of 53 drugs. In this work, we only considered gene/drug pairs whose CPIC evidence levels are class A (for which there is a sufficient level of evidence to suggest a recommendation) (Appendix A). Although the CPIC recommendations refer to a larger number of genes, as detailed in [https://cpicpgx.org/genes-drugs/ (accessed on 16 August 2022)], this work relies only on the 14 genes considered by PGxPOP at the time the analysis described here was carried out.

Frequencies in Sardinia of star allele haplotypes and diplotypes, and estimated prevalence of non-typical response phenotypes were compared with the corresponding values in UKBiobank populations reported by McInnes et al. [21] (Appendix A), where the burden of nontypical response phenotypes for each individual is estimated by counting the number of diplotypes with predicted nontypical response phenotypes across 14 genes with phenotypes. Gene phenotypes were determined to have a nontypical response if any CPIC guidance recommended an alternate dosage or drug for that phenotype.

Structural variants (SVs) were not called for any of the considered genes.

#### 4.3.2. Allele Frequency Analysis of PGx Actionable Variants from PharmGKB

The full list of PharmGKB [6] clinically relevant variants assigned to evidence classes 1A, 1B, 2A and 2B, downloaded from the PharmGKB website ([https://www.pharmgkb.org/], was accessed on 22 March 2022). Level 1A clinical annotations are typical of those variant-drug combinations characterized by the existence of precise variant-specific prescriptive guidance, such as in an FDA-approved drug label. Level 1B clinical annotations describe variant-drug combinations with a high level of evidence supporting the association but no specific prescriptive guidance, as was required for the previous annotations. Level 2A variants are in PharmGKB’s Tier 1 Very Important Pharmacogenes (VIPs), i.e., in known pharmacogenes, and, thus, with high probability of phenotype causation. Variants in Tier 2B clinical annotations are not present in PharmGKB Tier 1 VIPs and describe variant-drug combinations with a moderate level of evidence supporting association. Variant-drug pairs with PharmGKB levels 1B, 2A and 2B must all be supported by at least two independent publications. For more details about the PharmGKB scoring system, readers can refer to the documentation available at https://www.pharmgkb.org/page/clinAnnLevels (accessed on 16 August 2022).

### 4.4. Statistical Analysis

All statistical analyses were performed using R Studio 2022.07.1 Build 554 (R Foundation for Statistical Computing, Vienna, Austria). Significant differences in phenotype frequencies between our cohort and UKBiobank European population were analyzed by Fisher’s exact test. The False Discovery Rate (FDR) proposed by Benjamin and Hochberg [36] was used to correct the multiple analyses. Results were considered statistically significant when the *p*-value was <0.05.

## 5. Conclusions

In this work, we have completed what is, to date, the most extensive characterization of pharmacogenetic variability in an Italian population, specifically that of Sardinia. The analysis of the prevalence of PGx risk variants presented here may stimulate initiatives to implement large-scale pharmacogenetic strategies in Italy.

The impact of pharmacogenetic variation on health is, thus, patently obvious; in a future analysis it may be useful to consider that the impact falls disproportionally with age and it may be stronger on the elderly. The effects of aging result first from the relatively high usage of pharmaceutical drugs in the elderly (https://www.kff.org/health-reform/issue-brief/data-note-prescription-drugs-and-older-adults/ (accessed on 22 March 2022); https://hpi.georgetown.edu/rxdrugs/ (accessed on 22 March 2022)) and second from the relatively greater sensitivity to drugs in the elderly, so that the dosages, which are determined on younger adults, are often excessive for them, increasing the risk of side effects.

Variants in pharmacogenes can then further exacerbate the problems. Concerning germ-line mutations, we have dealt here with the effect of single variants on single drugs, but we know little about the effect of genetics on combinations of multiple drugs. This is again especially important for the elderly, who, compared to the middle-aged individuals who normally participate in clinical trials, are more frequently simultaneously prescribed multiple drugs (often in a ‘therapeutic cascade’, in which the side effects of one drug are treated with another drug). Correspondingly, additive or synergistic incidence of side effects and intensification of genetic variant effects can be expected. Furthermore, with age each individual accumulates new somatic mutations, and the liver—in which the majority of the genes relevant to ADME-Tox are expressed—is one of the tissues most exposed to environmental mutagens. It, therefore, tends to accumulate somatic mutations that can potentially further alter the function of pharmacogenes [37,38], although the extent to which this occurs has not been quantified. Further analyses should define more precisely the relative load of genetic risk as a function of age and the measures, like lower doses and substitution of drugs by others with different genetic risk profiles, that may mitigate it.

There are some limitations to this study that could be met in the future. First, only by using high coverage sequencing data on the exome or genome will it be possible to define with certainty the existence and prevalence of rarer variants with important effects specific to the Sardinian population. Second, we did not assess structural variants, but for example, *CYP2D6* is well-known to have structural variants including copy number variability and gene rearrangements between *CYP2D7*-*CYP2D6* known as *hybrid tandems*. Third, our analysis was limited to 14 genes, that could be analyzed with PGxPOP at the time this work was prepared; this limitation could be overcome by future analysis, that uses new available information on drug-gene pairs (unfortunately not implemented in PGxPOP). Two other limitations currently preclude estimation of the potential economic cost of non-stratification of patients based on genetic characteristics. The absence of personal data on drug prescriptions (which would allow us to understand how many people, and which individuals, are really at risk of adverse reactions to the drugs they use) and global data on consumption in Sardinia (which would allow us to make pharmacoeconomic estimates). 

Nevertheless, the findings demonstrate the value of characterizing allele frequencies in diverse populations and highlights the need for more PGx research on understudied populations, an important step in the corresponding refined implementation of modern personalized medicine.

## Figures and Tables

**Figure 1 ijms-23-10058-f001:**
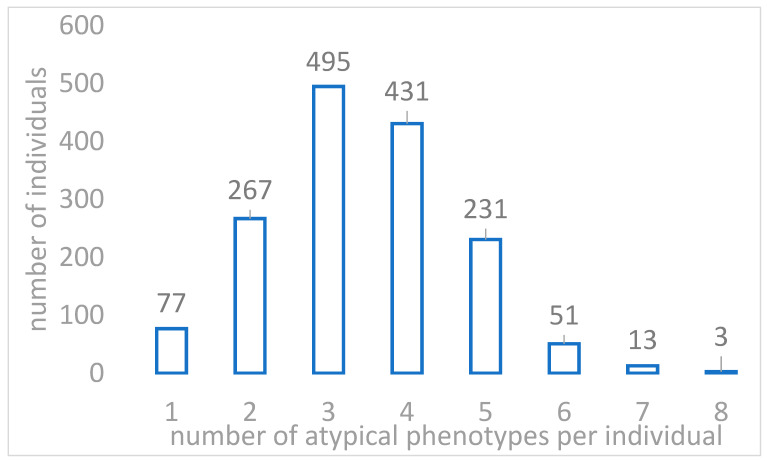
The figure shows the distribution of the number of abnormal drug response phenotypes detected in the 1577 individuals in the Sardinia cohort by PGxPop.

**Table 1 ijms-23-10058-t001:** The table shows how many individuals are at risk of an atypical drug response, in descending order of number of drugs.

Number of Individuals	Number of Drugs for Which an Atypical Response Is Expected
**1**	39
**3**	38
**6**	37
**12**	36
**16**	35
**14**	34
**5**	33
**7**	32
**5**	31
**19**	30
**38**	29
**57**	28
**47**	27
**49**	26
**40**	25
**45**	24
**43**	23
**51**	22
**36**	21
**34**	20
**42**	19
**92**	18
**132**	17
**125**	16
**127**	15
**89**	14
**58**	13
**58**	12
**22**	11
**22**	10
**12**	9
**11**	8
**2**	7
**9**	6
**27**	5
**49**	4
**68**	3
**55**	2
**40**	1
** *TOT = 1568 (99.43%)* **	

**Table 2 ijms-23-10058-t002:** Predicted non-typical response phenotype across the 14 pharmacogenes analyzed in the Sardinia cohort; each cell contains the number of individuals in the cohort to whom atypical response phenotypes (highlighted in **bold**) are assigned by PGxPop, divided by pharmacogene.

*Phenotypes*	*Pharmacogenes*													
	*CFTR*	*CYP2B6*	*CYP2C19*	*CYP2C9*	*CYP2D6*	*CYP3A5*	*CYP4F2*	*DPYD*	*IFNL3*	*NUDT15*	*SLCO1B1*	*TPMT*	*UGT1A1*	*VKORC1*
Decreased Function											**324**			
Decreased warfarin dose														**414**
Favorable response genotype									721					
Increased dose phenotype							**268**							
Indeterminate		50			207		11				133	2		
Intermediate Function												**101**		
Intermediate Metabolizer		**552**	**450**	**606**		**304**		**117**		2				
Ivacaftor irrelevant	1555													
Ivacaftor non-responsive	14													
Ivacaftor responsive	8													
Normal dose phenotype							534							
Normal Function											533	1472		
Normal Metabolizer		862	715	925	622	16		1460		1575			815	
Normal warfarin dose														385
Not available		8		2	748	14	764				201		762	
Poor Function											**51**	**2**		
Poor Metabolizer		**85**	**32**	**44**		**1243**								
Possible increased function											335			
Possibly decreased warfarin dose														**778**
Rapid Metabolizer		**20**	**343**											
Ultrarapid Metabolizer			**37**											
Unfavorable response genotype									**856**					
*Total non-typical phenotypes*	**14**	**552**	**519**	**650**	**0**	**1547**	**268**	**117**	**856**	**0**	**375**	**103**	**0**	**1192**

**Table 3 ijms-23-10058-t003:** For each gene the percentage of individuals in the Sardinia cohort who were at risk of atypical response is reported. Only Gene/Drug pairs with CPIC evidence of level A were considered here (according to https://cpicpgx.org/genes-drugs/ (accessed on 16 August 2022)). Estimated frequency of non-typical response phenotypes in Sardinians (SARD) were compared with the corresponding values in UKBiobank European populations (UKBB-EUR) reported by McInnes et al. [16] and differences among the two values are reported in column “Delta freq.” (See extended data in Appendix A). The *p*-value refers to pairwise comparison of phenotype frequencies in Sardinian and European populations. CYP2D6 predictions were omitted because frequencies were likely to be affected by copy number variants that are not considered here.

Gene	Related Drugs	Non-Typical Response Phenotypes	N SARD	Freq SARD	Delta Freq.	*p*-Value
** *CFTR* **	Ivacaftor	ivacaftor irrelevant	1555	0.98604946	0.02922937	7.77 × 10^−8^
		ivacaftor non-responsive	14	0.00887762	−0.0240865	5.38 × 10^−7^
		ivacaftor responsive	8	0.00507292	−0.00509874	0.10
		Not available	0	0	0	1
** *CYP2B6* **	efavirenz	Rapid Metabolizer	20	0.01268231	0.01235134	3.82 × 10^−65^
		Indeterminate	50	0.03170577	−0.01226846	0.04
		Intermediate Metabolizer	552	0.35003171	0.00655614	0.79
		Poor Metabolizer	85	0.05389981	−0.00298206	0.83
		Not available	8	0.00507292	−0.00231863	0.54
		Normal Metabolizer	862	0.54660748	−0.00122801	1
		Ultrarapid Metabolizer	0	0	−0.00011032	1
** *CYP2C19* **	amitriptyline, citalopram, sertraline, clopidogrel, escitalopram, imipramine, clomipramine, doxepin, trimipramine, voriconazole	Normal Metabolizer	715	0.45339252	0.05749207	1.63 × 10^−5^
		Rapid Metabolizer	343	0.21750159	−0.05355882	9.34 × 10^−6^
		Intermediate Metabolizer	450	0.28535193	0.02347028	0.07
		Ultrarapid Metabolizer	37	0.02346227	−0.02161518	0.00
		Poor Metabolizer	32	0.02029169	−0.00393495	0.54
		Not available	0	0	−0.00145625	0.39
		Indeterminate	0	0	−0.00019858	1
		Likely Intermediate Metabolizer	0	0	−0.00019858	1
		Likely Poor Metabolizer	0	0	0	-
** *CYP2C9* **	ibuprofen, piroxicam, celecoxib, meloxicam, phenytoin, flurbiprofen, tenoxicam, lornoxicam, warfarin	Normal Metabolizer	925	0.58655675	−0.05549347	2.20 × 10^−5^
		Intermediate Metabolizer	606	0.38427394	0.04975649	0.00
		Poor Metabolizer	44	0.02790108	0.00590293	0.23
		Indeterminate	0	0	−0.00028684	1
		Not available	2	0.00126823	0.00012088	1
** *CYP3A5* **	tacrolimus	Poor Metabolizer	1243	0.78820545	−0.08090889	1.09 × 10^−19^
		Intermediate Metabolizer	304	0.19277108	0.06775453	1.17 × 10^−14^
		Normal Metabolizer	16	0.01014585	0.0042988	0.078017
		Indeterminate	0	0	0	1
** *CYP4F2* **	warfarin	Not available	764	0.48446417	0.18778706	4.30 × 10^−56^
		Normal dose phenotype	534	0.33861763	−0.15050465	5.15 × 10^−31^
		Increased dose phenotype	268	0.16994293	−0.04138931	2.07 × 10^−4^
		Indeterminate	11	0.00697527	0.00410691	0.02
** *DPYD* **	fluorouracil, capecitabine	Intermediate Metabolizer	117	0.0741915	0.00471089	0.68
		Normal Metabolizer	1460	0.9258085	−0.0040269	0.76
		Poor Metabolizer	0	0	−0.00046335	1
		Not available	0	0	−0.00022064	1
** *IFNL3* **	ribavirin, peginterferon alfa-2a, peginterferon alfa-2b	Unfavorable response genotype	856	0.54280279	0.03075566	0.03
		Favorable response genotype	721	0.45719721	−0.03075566	0.03
** *NUDT15* **	azathioprine, mercaptopurine, thioguanine	Normal Metabolizer	1575	0.99873177	0.01104367	2.84 × 10^−04^
		Intermediate Metabolizer	0	0	−0.00849477	9.58 × 10^−4^
		Indeterminate	2	0.00126823	−0.00164426	0.52
		Not available	0	0	−0.00083844	0.68
		Poor Metabolizer	0	0	0	1
		Possible Intermediate Metabolizer	0	0	0	1
** *SLCO1B1* **	fluvastatin, lovastatin, pitavastatin, pravastatin, rosuvastatin, simvastatin	Normal Function	533	0.33798351	−0.22189911	1.25 × 10^−66^
		Possible Decreased Function	335	0.21242866	0	0
		Possible Increased Function	0	0	−0.08680111	2.87 × 10^−33^
		Not available	201	0.1274572	0.07320099	7.12 × 10^−34^
		Indeterminate	133	0.08433735	0.02026251	3.54 × 10^−3^
		Poor Function	51	0.03233989	0.01027555	0.02
		Decreased Function	324	0.20545339	−0.00737923	0.68
		Possible Poor Function	0	0	0	NA
** *TPMT* **	azathioprine, mercaptopurine, thioguanine	Normal Function	1472	0.93341788	0.0372306	7.17 × 10^−6^
		Intermediate Function	101	0.06404566	−0.03380969	2.74 × 10^−5^
		Indeterminate	2	0.00126823	−0.00173252	0.50
		Poor Function	2	0.00126823	−0.00144568	0.58
		Possible Intermediate Metabolizer	0	0	−0.00017651	1
		Not available	0	0	0	1
** *UGT1A1* **	atazanavir, irinotecan	Normal Metabolizer	815	0.51680406	0.05715974	2.50 × 10^−5^
		Intermediate Metabolizer	0	0	−0.04417281	1.10 × 10^−16^
		Not available	762	0.48319594	−0.00903741	0.67
		Poor Metabolizer	0	0	−0.00366268	0.05
		Indeterminate	0	0	−0.00028684	1
** *VKORC1* **	warfarin	Normal warfarin dose	385	0.24413443	−0.15075106	1.39 × 10^−32^
		Decreased warfarin dose	414	0.26252378	0.12201807	1.24 × 10^−40^
		Possibly decreased warfarin dose	778	0.49334179	0.02873299	0.05

**Table 4 ijms-23-10058-t004:** Summary of the variants associated with levels of evidence (1A, 1B, 2A or 2B) and with absolute difference in allele frequency between Sardinians and Europeans ≥ 5%. Abbreviations: BP = Position; RSID = Reference SNP cluster identifier; CHR = Chromosome; A2 = ALT allele.

**CHR**	**BP**	**RSID**	**Gene**	**A2**	**SARD_A2_FRQ**	**ALFA_A2_FRQ**	**Delta**
**1**	161514542	rs396991	*FCGR3A*	C	0.528	0.317	0.211
**16**	31104509	rs8050894	*VKORC1*	G	0.523	0.374	0.148
**19**	15990431	rs2108622	*CYP4F2*	T	0.415	0.294	0.120
**16**	31107689	rs9923231	*VKORC1*	T	0.510	0.392	0.117
**16**	31104878	rs9934438	*VKORC1*	A	0.510	0.395	0.114
**6**	39325078	rs20455	*KIF6*	G	0.463	0.364	0.099
**1**	11856378	rs1801133	*MTHFR*	A	0.424	0.349	0.075
**16**	31105554	rs2884737	*VKORC1*	C	0.353	0.278	0.074
**1**	97981395	rs1801159	*DPYD*	C	0.252	0.199	0.053
**CHR**	**BP**	**RSID**	**Gen**	**A2**	**SARD_A2_FRQ**	**ALFA_EUR_FRQ**	**Delta**
**1**	98348885	rs1801265	*DPYD*	G	0.159	0.215	−0.056
**21**	46957794	rs1051266	*SLC19A1*	C	0.511	0.568	−0.057
**6**	31543031	rs1800629	*TNF*	A	0.050	0.159	−0.109
**16**	31103796	rs2359612	*VKORC1*	G	0.490	0.604	−0.114

**Table 5 ijms-23-10058-t005:** Clinical annotations of the variants associated with levels of evidence (1A, 1B, 2A or 2B) and with absolute difference in allele frequency between Sardinians and Europeans ≥ 5%. Pediatric population column: 1 = yes. Data were obtained from the PharmGKB website [17].

Clinical Annotation ID	Variant/Haplotypes	is_top	Gene	Level of Evidence	Phenotype Category	Drug(s)	Phenotype(s)	Pediatric Population
**1184661194**	rs2108622	rs2108622	*CYP4F2*	2A	Dosage	acenocoumarol	Atrial Fibrillation	0
**981204044**	rs9923231	rs9923231	*VKORC1*	1A	Dosage	acenocoumarol		0
**1183704228**	rs9934438	rs9934438	*VKORC1*	2A	Dosage	acenocoumarol		0
**1451237940**	rs9923231	rs9923231	*VKORC1*	1A	Dosage	phenprocoumon		1
**1451244040**	rs9934438	rs9934438	*VKORC1*	2A	Dosage	phenprocoumon		1
**655385400**	rs2108622	rs2108622	*CYP4F2*	1A	Dosage	warfarin		1
**982035703**	rs2884737	rs2884737	*VKORC1*	2A	Dosage	warfarin		0
**655385392**	rs9934438	rs9934438	*VKORC1*	1B	Dosage	warfarin		1
**655385028**	rs8050894	rs8050894	*VKORC1*	1B	Dosage	warfarin		0
**655385012**	rs9923231	rs9923231	*VKORC1*	1A	Dosage	warfarin		1
**655385024**	rs2359612	rs2359612	*VKORC1*	1B	Dosage	warfarin		0
**655384799**	rs1800629	rs1800629	*TNF*	2B	Efficacy	etanercept	Arthritis, Psoriatic; Arthritis, Rheumatoid; Crohn Disease; Inflammation; Psoriasis; Spondylitis, Ankylosing	0
**1451245360**	rs1051266	rs1051266	*SLC19A1*	2A	Efficacy	methotrexate	Arthritis, Rheumatoid	0
**655384621**	rs20455	rs20455	*KIF6*	2B	Efficacy	pravastatin	Coronary Disease; Myocardial Infarction	0
**1444608384**	rs396991	rs396991	*FCGR3A*	2B	Efficacy	rituximab	Arthritis, Rheumatoid; Neuromyelitis Optica	0
**1447672998**	rs9923231	rs9923231	*VKORC1*	2A	Efficacy	warfarin	time to therapeutic inr	1
**1447673015**	rs9923231	rs9923231	*VKORC1*	2A	Efficacy	warfarin	time in therapeutic range	1
**1451286320**	rs1801159	rs1801159	*DPYD*	1A	Toxicity	capecitabine	Neoplasms	0
**1451287240**	rs1801265	rs1801265	*DPYD*	1A	Toxicity	capecitabine	Neoplasms	0
**981201981**	rs1801265	rs1801265	*DPYD*	1A	Toxicity	fluorouracil	Neoplasms	1
**981201962**	rs1801159	rs1801159	*DPYD*	1A	Toxicity	fluorouracil	Neoplasms	0
**827848365**	rs1801133	rs1801133	*MTHFR*	2A	Toxicity	methotrexate	Drug Toxicity;hematotoxicity; Leukopenia; Lymphoma; mucositis; Neoplasms; Neutropenia; Osteosarcoma; Precursor Cell Lymphoblastic Leukemia-Lymphoma; primary central nervous system lymphoma; Thrombocytopenia; Toxic liver disease	1
**655385307**	rs1801133	rs1801133	*MTHFR*	2A	Toxicity	methotrexate	Arthritis, Juvenile Rheumatoid; Arthritis, Psoriatic; Arthritis, Rheumatoid; Drug Toxicity	1
**1451243676**	rs9923231	rs9923231	*VKORC1*	2A	Toxicity	phenprocoumon	Hemorrhage;over-anticoagulation; time above therapeutic range	0
**1449269910**	rs9923231	rs9923231	*VKORC1*	2A	Toxicity	warfarin	Hemorrhage	1
**1447673005**	rs9923231	rs9923231	*VKORC1*	1B	Toxicity	warfarin	over-anticoagulation	1

**Table 6 ijms-23-10058-t006:** Summary of the variants associated with lower levels of evidence (3 or 4) and with differences in allele frequency between Sardinians and Europeans ≥ 20%. Abbreviations: POS = position; RSID = Reference SNP cluster identifier; CHR = Chromosome; A2 = ALT allele.

CHR	POS	RSID	Gene	A2	SARD_A2_FRQ	ALFA_A2_FRQ	Delta
**1**	1:207753621	rs2274567	*CR1*	G	0.608	0.195	0.413
**6**	6:31093482	rs3131003	*PSORS1C1*	A	0.739	0.432	0.308
**6**	6:31093587	rs3815087	*PSORS1C1*	A	0.507	0.217	0.290
**13**	12:13953118	rs2058878	*GRIN2B*	A	0.602	0.338	0.265
**16**	12:85243681	rs6539870	*IL1B*	G	0.486	0.221	0.264
**4**	3:45732515	rs2742421	*SACM1L*	G	0.687	0.427	0.260
**6**	6:31603770	rs11229	*PRRC2A*	G	0.435	0.176	0.259
**15**	15:30193316	rs813676	*TJP1*	C	0.761	0.504	0.257
**6**	6:31107361	rs2233945	*PSORS1C1*	A	0.417	0.163	0.254
**6**	6:31604591	rs10885	*PRRC2A*	T	0.436	0.182	0.254
**6**	6:31018546	rs2523864	*HCG22*	T	0.679	0.433	0.245
**19**	19:15959200	rs2189784	*CYP4F2*	A	0.667	0.423	0.245
**6**	6:31022113	rs3873352	*HCG22*	G	0.336	0.091	0.245
**6**	6:43737794	rs13207351	*VEGFA*	G	0.611	0.367	0.244
**3**	3:151090996	rs9859552	*P2RY12*	T	0.338	0.094	0.244
**6**	6:31543101	rs361525	*TNF*	A	0.292	0.054	0.238
**6**	6:31542308	rs1799964	*TNF*	C	0.447	0.214	0.233
**7**	6:78173281	rs130058	*HTR1B*	A	0.414	0.183	0.231
**5**	5:158750769	rs3213094	*IL12B*	T	0.427	0.206	0.221
**7**	7:86331756	rs2189814	*GRM3*	C	0.375	0.159	0.216
**11**	10:32202069	rs2799018	*ARHGAP12*	T	0.611	0.396	0.215
**1**	1:161514542	rs396991	*FCGR3A*	C	0.528	0.317	0.211
**11**	10:92619161	rs7905446	*HTR7, RPP30*	G	0.480	0.276	0.205
**6**	6:33047612	rs3097671	*HLA-DPB1*	C	0.366	0.163	0.203
**10**	1:29161999	rs2236855	*OPRD1*	A	0.386	0.185	0.201

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
