# Peer review of "Genetic Variation among Pharmacogenes in the Sardinian Population"

_ijms, 2022, doi:10.3390/ijms231710058_

Round 1
Author Response
REVIEWER 1
Congratulations for your work, I really found it very interesting and, as you comment, this kind of studies underline the importance of the pharmacogenomics characterization considering specific populations, this will allow to develop population-specific gene panels to help in the implementation of real personalized medicine.
On the other hand, I found many aspects that may be improved. I hope my comments may help to improve your manuscript.
[AU] We thank the reviewer for the positive and constructive feedback for our work.
Please, use appropriate nomenclature for pharmacogenetic terms. https://www.nature.com/articles/gim201687. E.g. Genes must be in italics, please revise throughout the text, tables included.
[AU] We thank the reviewer to point this out. All gene names were converted to italics. A reference to https://www.nature.com/articles/gim201687 was added.
It may be useful to highlight in the introduction the influence of this kind of study on the clinical implementation of pharmacogenetics. Reporting different implementation strategies that are being developed (e.g. PMID: 28044932), the increasing number of drug-gene interactions being found, and novel research projects and drug-gene association studies (e.g. PMID: 35250581 PMID: 34834388, PMID: 35832565).
[AU] We thank the reviewer for this suggestion. Indeed, the following sentence was added to the introduction: “The analysis of the prevalence of PGx-risk variants in target populations, in combination with actual data on drug usage, makes possible to predict the proportion of the population for which genetics could lead therapy decision. Additionally, i) the analysis of PGx variants prevalence, ii) the results of clinical trials evaluating patient outcomes and cost-effectiveness of PGx-markers [PMID 35819423] and iii) outcomes of implementation strategies [PMID: 28044932 and 28027596], could support a coordinate pharmacogenetic program in the European healthcare systems.”
Please, revise the layout in table 2 (e.g. First column at the left, first row in a single line etc.). It would help to understand it.
[AU] We thank the reviewer for this suggestion. To increase the readability of the table, we used banded columns and highlighted in bold the names of atypical response phenotypes in the first column.
Dear authors, I´m sorry but I don´t really understand “figure 2”. It means that 56.12% of Sardinians have non-typical response to all those drugs altogether (clopidogrel, statins…), or you mean to at least one of those drugs? In my opinion it does not provide useful information, you had just commented that 99.43% of patients had a potential non-typical response to drugs related to studied genes. Following with this, I think this figure might be improved if changed by a table reporting studied drug-gene interactions and resulting phenotypes. I know that you included this information as supplementary material, but it would be very useful. This way we can differentiate those drugs related to more than one drug (as warfarin). Please, see the following table as an example to let you understand what I mean, but feel free to improve it as you consider necessary.
Gene |
Related Drugs |
Non-typical response phenotypes |
N (%) Your population |
N (%) compared population |
p-value (diference between populations) |
CYP2C19 |
Clopidogrel, citalopram, escitalopram… |
IM |
|
|
|
PM |
|
|
|
||
UM |
|
|
|
||
RM |
|
|
|
||
EM (normal) |
|
|
|
||
SLCO1B1 |
Fluvastatin, rosuvastatin |
|
|
|
|
… |
|
|
|
|
|
[AU] We thank the reviewer for the helpful suggestion. We removed the Figure 2 and replaced it with a new table, Table 3.
In tables 3 and 5, please report the gene for each rs. It would be helpful for reading.
[AU] We agreed with the reviewer that adding the gene name is helpful for reading. Accordingly, we added a column with gene names in both tables 4 (previously Table 3) and 6 (previously Table 5).
In table 4, please include the reference in the title, reporting that you obtained the data from the PharmGKB.
[AU] We thank the reviewer for this suggestion. We added the following sentence in the title of Table 5 (prev. Table 4): “Data were obtained from the PharmGKB website [17].” in the title of Table 5 (prev. Table 4).
Dear authors, could you provide the inclusion/exclusion criteria, even in a single section in the materials and methods, explaining how you chose those fourteen clinically relevant genes, drug-gene interactions, rs, and not others? I mean, did you choose all drug-gene interactions reported with level of evidence 1A-1B?
[AU] We thank the reviewer for rising this point. One of the objectives of this work was to obtain from the Sardinian sequence dataset the same data calculated by McInnes et al on the UKBiobank population. For this reason, we used the same tool (PGxPOP) used by McInnes et al. Thus, the choice of genes is solely related to the characteristics of PGxPOP, which at the time of use only allowed analysis on 14 genes.
We have made it clearer in the methods with this sentence (section “3.3.1. - Haplotype and phenotype calling”):
“Although the CPIC recommendations refer to a larger number of genes, as detailed in https://cpicpgx.org/genes-drugs/, this work relies only on the 14 genes considered by PGxPOP at the time the analysis described here was carried out..”
I know you comment aspects related to this throughout the manuscript, even you explain it about CYP2D6 in the discussion, but it is difficult to find this information. If there is not an impartial criterion, please, include it as a limitation.
[AU] We thank the reviewer for pointing this out. We added the following sentence to the discussion to point out that this is a limitation explained by the PGxPOP capabilities:
Third, our analysis is limited to 14 genes, that could be analyzed with PGxPOP at the time this work was prepared; this limitation could be overcome by future analysis, that will use new available information on drug-gene pairs (unfortunately not implemented in PGxPOP).

Reviewer 2 Report
I'm extremely surprised that only CYP2D6 or UGT1A1 normal metabolizers were identified in this population (the others being indeterminate or not available). Looking at the data for CYP2D6 in Supplementary Table 4, the low diversity of variants called is striking, with *119 being relatively highly represented. This is even more emphasized when you compare this data to other published allele frequencies from Sardinian populations (e.g. PMID 15340360 for CYP2D6).
Have the authors verified that there were no issues with the allele calling process and, if these are indeed the true allele frequencies, are they able to comment on why this is the case? For CYP2D6, why is the number of 'not available' phenotypes so high and can the authors identify reasons that so many diplotypes were assigned to this group? I understand that PGxPOP is unable to identify structural variants, which would mean that CYP2D6 ultrarapid metabolizers would not be identified, but not finding any intermediate metabolizers for either gene is very unexpected, especially when compared to the UK Biobank data.
Some more minor comments:
Line 47: While adverse reactions are a relatively common result of drug treatment, I disagree with the sentiment that drug treatment can be characterized by ADRs.
Line 55: The mention of ethnic groups is unnecessary here and I suggest removing it. While there is data showing differences in drug response between different races/ethnicities, it is highly unlikely that race/ethnicity is itself the determining factor in the difference in response and more likely that it is a proxy marker for unidentified causative genetic variants, which are not necessarily limited to people of that particular race/ethnicity.
Line 68: Phenotypes are assigned to combinations of haplotypes regardless of how many SNPs each haplotype contains. The CYP2C9*2 and *3 haplotypes, for example, are defined by the presence of single SNPs.
Line 72: PharmGKB is short for the Pharmacogenomics Knowledgebase. Please also follow the citation policy given on the website.
Line 94: Please provide an appropriate citation for CPIC.
Line 199: It should be noted that rs9934438 is in high LD with rs9923231.
Line 294: Variant-drug pairs with PharmGKB levels 1B, 2A and 2B must all be supported by at least two independent publications.
Author Response
REVIEWER 2
I'm extremely surprised that only CYP2D6 or UGT1A1 normal metabolizers were identified in this population (the others being indeterminate or not available). Looking at the data for CYP2D6 in Supplementary Table 4, the low diversity of variants called is striking, with *119 being relatively highly represented. This is even more emphasized when you compare this data to other published allele frequencies from Sardinian populations (e.g. PMID 15340360 for CYP2D6).
Have the authors verified that there were no issues with the allele calling process and, if these are indeed the true allele frequencies, are they able to comment on why this is the case? For CYP2D6, why is the number of 'not available' phenotypes so high and can the authors identify reasons that so many diplotypes were assigned to this group? I understand that PGxPOP is unable to identify structural variants, which would mean that CYP2D6 ultrarapid metabolizers would not be identified, but not finding any intermediate metabolizers for either gene is very unexpected, especially when compared to the UK Biobank data.
[AU] We thank the reviewer for the comment. Unfortunately, with our current genetic map, successfully used in different in many different publications (see Orrù V et al., Nat Gen 2020; Sidore et al Nat Genet. 2015; Orru et al, Cell 2013; Steri et al, NEJM 2017; Zoledziewska M et al, Nat Genet 2015) is not possible to perform a reliable prediction of CYP2D6 alleles. Indeed, genetic map has been obtained by array data with low coverage sequencing not sufficient to account the complete variability in CYP2D6 gene, for which the impact of structural variants is extremely important. We therefore performed a preliminary analysis with 65 deeply sequenced samples from the same cohort (mean coverage > 30x); we used the Aldy tool [https://www.nature.com/articles/s41467-018-03273-1] to call the CYP2D6 star alleles in these 65 samples, results are listed in the Table below:
haplotype |
European haplotype frequency (from McInnes et al) |
Sard haplotype frequency |
*1 |
0,369313 |
0,269231 |
*10 |
0,010591 |
0,069231 |
*117 |
0,000883 |
0,007692 |
*119 |
0,000066 |
0,007692 |
*139 |
0,000088 |
0,038462 |
*2 |
0,168605 |
0,176923 |
*3 |
0,017420 |
0,015385 |
*34 |
0,000099 |
0,015385 |
*35 |
0,048928 |
0,046154 |
*4 |
0,156502 |
0,061538 |
*41 |
0,093255 |
0,100000 |
*65 |
0,000022 |
0,007692 |
Other |
0,134228 |
0,184615 |
In our opinion, the results are in line with those reported by McInnes et al in Supplementary File 1. We decided to not include them in our manuscript because the number of samples (N=65) used for this analysis is very low.
Regarding UGT1A1, we checked again the raw data and confirmed the results reported in the submitted version of the paper. As the Reviewer can see from the new Table 3, the main difference with UKBiobank predictions (McInnes et al) refers to the phenotype “Intermediate Metabolizer” (0 in Sardinians, 4% in UKBB Europeans). Furthermore, we performed haplotype calling of UGT1A1 with Aldy tool on the deep coverage samples (N=65) and phenotype definition according to https://www.pharmgkb.org/page/ugt1a1RefMaterials, and we obtained the following results:
- Normal Metabolizer (*1 / *1): 37/65 (57%);
- Indeterminate phenotype (*1 / *80 or *80 / *80): 28/65 (43%);
- Other: 0.
Although the number of deep seq samples is low, the results support what observed with PGxPOP on the entire cohort.
We added this sentence to the Results section:
To understand whether the unexpected result for CYP2D6 is due to a limitation of the SardiNIA genetic map in the CYP2D6 region, we performed an exploratory analysis with 65 deeply sequenced samples from the same cohort (mean coverage > 30x, data not shown here), where CYP2D6 star alleles were called with the Aldy tool [https://www.nature.com/articles/s41467-018-03273-1]; results show that common CYP2D6 star alleles have comparable frequencies to those reported for the European samples in UKBiobank cohort (McInnes et al REF).
Some more minor comments:
Line 47: While adverse reactions are a relatively common result of drug treatment, I disagree with the sentiment that drug treatment can be characterized by ADRs.
[AU] We thank the reviewer to rise this point. Sentence has been changed to reduce the emphasis on the fact that “drug treatment can be characterized by ADRs”.
Line 55: The mention of ethnic groups is unnecessary here and I suggest removing it. While there is data showing differences in drug response between different races/ethnicities, it is highly unlikely that race/ethnicity is itself the determining factor in the difference in response and more likely that it is a proxy marker for unidentified causative genetic variants, which are not necessarily limited to people of that particular race/ethnicity.
[AU] The reference to race/ethnicity has been removed from row 55.
Line 68: Phenotypes are assigned to combinations of haplotypes regardless of how many SNPs each haplotype contains. The CYP2C9*2 and *3 haplotypes, for example, are defined by the presence of single SNPs.
[AU] We agree with the reviewer and removed the sentence referring to a single, specific SNPs for phenotypes definition.
Line 72: PharmGKB is short for the Pharmacogenomics Knowledgebase. Please also follow the citation policy given on the website.
[AU] We thank the reviewer to underline this point. We changed the long name and added two references according to the citation policy of the PharmGKB website
Line 94: Please provide an appropriate citation for CPIC.
[AU] Citation has been added
Line 199: It should be noted that rs9934438 is in high LD with rs9923231.
[AU] We thank the reviewer to point this out. We added the following sentence to the Results section: It should be noted that SNPs rs9934438 and rs9923231 are in high LD (rs9934438 and rs9923231 are in high LD, with r2=1 in both Sardinians and Europeans).
Line 294: Variant-drug pairs with PharmGKB levels 1B, 2A and 2B must all be supported by at least two independent publications.
[AU] Thank you for the suggestion. Information has been added at the end of the section “Allele frequency analysis of PGx actionable variants from PharmGKB (Methods)” that was accordingly slightly modified as follows:
The full list of PharmGKB [19] clinically relevant variants assigned to evidence classes 1A, 1B, 2A and 2B, downloaded from the PharmGKB website ([https://www.pharmgkb.org/], accessed March 2022). Level 1A clinical annotations are typical of those variant-drug combinations characterized by the existence of precise variant-specific prescriptive guidance, such as in an FDA-approved drug label.
Level 1B clinical annotations describe variant-drug combinations with a high level of evidence supporting the association but no specific prescriptive guidance, as was required for the previous annotations. Level 2A variants are in PharmGKB's Tier 1 Very Important Pharmacogenes (VIPs), i.e., in known pharmacogens, and thus with high probability of phenotype causation. Variants in Tier 2B clinical annotations are not present in PharmGKB Tier 1 VIPs and describe variant-drug combinations with a moderate level of evidence supporting association. Variant-drug pairs with PharmGKB levels 1B, 2A and 2B must all be supported by at least two independent publications.
For more details about the PharmGKB scoring system, readers can refer to the documentation available at https://www.pharmgkb.org/page/clinAnnLevels.

Round 2
Reviewer 2 Report
Thank you for addressing my previous comments. I'm satisfied that this paper is now acceptable for publication.